Retinoic Acid Induced Protein 14 (Rai14) is dispensable for mouse spermatogenesis

Wu Yangyang 1
Wang Ting 2
Zhao Zigao 3
Liu Siyu 1
Shen Cong 2
Li Hong 2
Liu Mingxi 1
Zheng Bo mansnoopy@163.com 2
Yu Jun 4
Huang Xiaoyan bbhxy@njmu.edu.cn 1
1 State Key Laboratory of Reproductive Medicine, Department of Histology and Embryology, Nanjing Medical University , Nanjing , China
2 Center for Reproduction and Genetics, Suzhou Municipal Hospital, the Affiliated Suzhou Hospital of Nanjing Medical University , Suzhou , China
3 Yunnan Institute of Population and Family Planning Science and Technology , Kunming , China
4 Institute of Reproductive Medicine, Medical School, Nantong University , Nantong , China
Yanowitz Judith
Electronic publication date: 2021 Feb 19
Publication date: 2021
Volume: 9
Electronic Location ID: e10847
Received 2020 Oct 1; Accepted 2021 Jan 6
Copyright: ©2021 Wu et al.
Copyright year: 2021
Copyright holder: Wu et al.
License: This is an open access article distributed under the terms of the Creative Commons Attribution License, which permits unrestricted use, distribution, reproduction and adaptation in any medium and for any purpose provided that it is properly attributed. For attribution, the original author(s), title, publication source (PeerJ) and either DOI or URL of the article must be cited.
License URL: https://creativecommons.org/licenses/by/4.0/

Keywords: Rai14, Spermatogenesis, Knockout, Ectoplasmic specialization, F-actin

Funding: National Natural Science Foundation of China 81901532 81901533 Natural Science Foundation of Jiangsu Province BK20190188 Suzhou Introduced Project of Clinical Medical Expert Team SZYJTD201708 Suzhou Key Laboratory of Male Reproduction Research SZS201718 This work was supported by the National Natural Science Foundation of China (81901532 and 81901533), the Natural Science Foundation of Jiangsu Province (BK20190188), the Suzhou Introduced Project of Clinical Medical Expert Team (SZYJTD201708) and the Suzhou Key Laboratory of Male Reproduction Research (SZS201718). The funders had no role in study design, data collection and analysis, decision to publish, or preparation of the manuscript.

==============================
Background

Retinoic Acid Induced Protein 14 (Rai14) is an evolutionarily conserved gene that is highly expressed in the testis. Previous experiments have reported that small interfering RNA (siRNA)-mediated gene knockdown (KD) of Rai14 in rat testis disrupted spermatid polarity and transport. Of note, a gene knockout (KO) model is considered the “gold standard” for in vivo assessment of crucial gene functions. Herein, we used CRISPR/Cas9-based gene editing to investigate the in vivo role of Rai14 in mouse testis.

Methods

Sperm concentration and motility were assayed using a computer-assisted sperm analysis (CASA) system. Histological and immunofluorescence (IF) staining and transmission electron microscopy (TEM) were used to visualize the effects of Rai14 KO in the testes and epididymides. Terminal deoxynucleotidyl transferase-dUTP nick-end labeling (TUNEL) was used to determine apoptotic cells. Gene transcript levels were calculated by real-time quantitative PCR.

Results

Rai14 KO in mice depicted normal fertility and complete spermatogenesis, which is in sharp contrast with the results reported previously in a Rai14 KD rat model. Sperm parameters and cellular apoptosis did not appear to differ between wild-type (WT) and KO group. Mechanistically, in contrast to the well-known role of Rai14 in modulating the dynamics of F-actin at the ectoplasmic specialization (ES) junction in the testis, morphological changes of ES junction exhibited no differences between Rai14 KO and WT testes. Moreover, the F-actin surrounded at the ES junction was also comparable between the two groups.

Conclusion

In summary, our study demonstrates that Rai14 is dispensable for mouse spermatogenesis and fertility. Although the results of this study were negative, the phenotypic information obtained herein provide an enhanced understanding of the role of Rai14 in the testis, and researchers may refer to these results to avoid conducting redundant experiments.

Introduction

Spermatogenesis is a complex process of germ cell proliferation and differentiation, and is associated with the extensive restructuring of cell junctions at the Sertoli-Sertoli cell and Sertoli-germ cell interfaces (Upadhyay et al., 2012). Of the various junctions in the seminiferous epithelium, the ectoplasmic specialization (ES) junction is a testis-specific adherens junction. It is an atypical actin-based junction at the blood–testis barrier (BTB) between adjacent Sertoli cells and is referred to as the basal ES, and between Sertoli cell and spermatid near the luminal surface of the seminiferous epithelium and is termed as the apical ES (Cheng & Mruk, 2010). The ES junction consists of hexagonal bundles of actin filaments sandwiched between cisternae of the endoplasmic reticulum and the plasma membranes (Lee & Cheng, 2004). During spermatogenesis, the ES primarily facilitates germ cell transport, polarity, and spermiation (Qian et al., 2014).

The Retinoic Acid Induced Protein 14 (RAI14) gene is a developmentally regulated gene that is induced by retinoic acid. RAI14 was originally identified in human retinal pigment epithelial cells (Kutty et al., 2001). In human tissues, RAI14 is predominantly expressed in the placenta and testes (Kutty et al., 2006a). The RAI14 protein comprises six ankyrin repeats and a long coiled-coil domain near the N-terminal region and the C-terminus, respectively. These domains are involved in protein-protein interactions (Kutty et al., 2006b). Qian et al. demonstrated that RAI14 is expressed at both the Sertoli and germ cells in rat testes (Qian et al., 2013b). They also demonstrated specific distribution of RAI14 at both the basal ES and the apical ES in rat testes. They found that RAI1 regulated F-actin organization at the ES. In another study, Qian et al. found that small interfering RNA (siRNA)-mediated Rai14 KD in Sertoli cells disturbed the permeability of the cell junction as well as disrupted F-actin in vitro. Moreover, siRNA-mediated Rai14 KD in rat testis in vivo disrupted spermatid polarity and adhesion as well as spermatid movement, which were caused by the disruption of the apical ES (Qian, Mruk & Cheng, 2013a).

RAI14 has also been found to be predominantly expressed in mouse testis (Kutty et al., 2006b). However, little is known about its function during mouse spermatogenesis. In the present study, we aimed to uncover the physiological role of RAI14 in mouse testis through CRISPR/Cas9-based gene editing.

Materials and Methods

Mice

CD-1 mice were obtained and maintained in a temperature and humidity-controlled room at the Experimental Animal Center of Nanjing Medical University with food and water provided ad libitum. Mice were randomly divided into cages. All individualized ventilated cages were capable of hosting 4–5 mice. Cages density, bedding, and sanitation frequency was similar in all cages. At the end of the study, mice were anesthetized with carbon dioxide. This study was carried out in strict accordance with the guidelines of the Institutional Animal Care and Use Committee of Nanjing Medical University (China). Animal use was approved by the Animal Ethical and Welfare Committee (AEWC) of Nanjing Medical University (Permit Number: IACUC-2004020). For the generation of Rai14 KO mice, Cas9 plasmid (Addgene, Watertown, MA, USA) was linearized and transcribed into mRNA in vitro using a T7 Transcription Kit (Ambion, Austin, TX, USA). The sgRNAs were designed based on exon 3 of Rai14. The target sgRNA sequence was 5′-CCGTCTGCTGCAGGCTGTGGAGA -3′ and 5′-GAGAAGGTGGCCTCACTGCTGGG -3′, respectively. Cas9 mRNA and sgRNA were microinjected into CD-1 mouse zygotes and transferred into the oviducts of pseudopregnant CD-1 females. The Rai14 genotype was verified by PCR amplification (Vazyme, Nanjing, China) using the following primers: (forward 5′- GGAGTTTGCTGATGGCTGGTATT-3′ and reverse 5′- CTCCATCGCCAACACTGTAAGAA-3′).

Western blot

Western blot analysis was performed according to our previously reported method with minor modifications (Shen et al., 2019; Zheng et al., 2018). Briefly, testis lysates were separated through electrophoresis and electro-transferred to polyvinylidene difluoride (PVDF) membranes (Bio-Rad, Hercules, USA). The PVDF membranes were blocked with 5% nonfat milk for 2 h at room temperature (RT) and incubated overnight at 4 °C with the primary antibodies: anti-RAI14 rabbit antibody (17507-1-AP; Proteintech, Chicago, IL, USA) at a dilution of 1:2,000 and anti-Tubulin mouse antibody (AT819; Beytime, Nantong, China) at a dilution of 1:20,000. Blots were then washed and incubated at RT for 2 h with horseradish peroxidase- conjugated secondary antibodies at a dilution of 1:2,000 (Thermo Scientific, Waltham, USA). The signals were visualized using enhanced chemiluminescent (Thermo Scientific, Waltham, USA).

Immunofluorescence

Testes were dissected and fixed in modified Davidson’s fluid (MDF) for at least 48 h before being embedded in paraffin. The sections were deparaffinized in xylene, hydrated in graded ethanol and boiled in sodium citrate buffer for antigen retrieval as previously described (Shen et al., 2018; Zhao et al., 2019). Sections were then blocked with 1% bovine serum albumin at RT for 2 h and incubated overnight at 4 °C with the primary antibodies: anti-RAI14 rabbit antibody (17507-1-AP; Proteintech, Chicago, IL, USA) at a dilution of 1:200, anti-Lin28 rabbit antibody (ab46020; Abcam, Cambridge, MA, USA) at a dilution of 1:500, anti-γH2AX mouse antibody (ab26350; Abcam, Cambridge, MA, USA) at a dilution of 1:1000, anti-Vimentin mouse antibody (sc-6260; Santa Cruz Biotechnology, Santa Cruz, CA, USA) at a dilution of 1:200, anti- HSD-3β (sc-515120; Santa Cruz Biotechnology, Santa Cruz, CA, USA) at a dilution of 1:500, anti-β-catenin mouse antibody (610153; BD Sciences, Franklin Lakes, NJ, USA) at a dilution of 1:200, anti-Espin (611656; BD Sciences, Franklin Lakes, NJ, USA) at a dilution of 1:300 and anti-Palladin rabbit antibody (10853-1-AP; Proteintech, Chicago, IL, USA) at a dilution of 1:400. Slides were rinsed before incubation with Alexa-Fluor secondary antibodies (Thermo Scientific, Waltham, USA) for 1 h at 37 °C. Finally, the slides were stained with Hoechst (Invitrogen, Carlsbad, CA, USA) and images were captured using a confocal microscope (Zeiss LSM800, Carl Zeiss, Oberkochen, Germany).

Fertility test

Adult males of Rai14 wild-type (WT, +/+) and knockout (KO, -/-) were housed individually with two WT CD-1 females for 16 weeks. The numbers of vaginal plugs and pups were counted, and the dates of birth were recorded in detail for each litter.

Computer-assisted sperm analysis (CASA)

Sperm were collected from the cauda epididymis and suspended in human tubal fluid and maintained at 37 °C. Sperm samples were then diluted, placed on an 80-µm chamber slide, and analyzed using Oval Head Toxicology software and the Hamilton Thorne’s Ceros II analyzer (Beverly, MA, USA). The parameters analyzed included sperm concentration and motility.

Histological analysis

The testes and epididymides were obtained from Rai14 WT and KO mice, fixed in MDF for at least 48 h, dehydrated in graded ethanol, embedded in paraffin, and finally sectioned into 5-µm thickness. After deparaffinization, the epididymis and testis slides were stained with hematoxylin and eosin (HE) and periodic acid Schiff (PAS) reagent, respectively. For electron microscopy analysis, testis and sperm were fixed in 4% and 2% glutaraldehyde, respectively. The samples were embedded in araldite and sectioned into 80-nm thickness. Images were examined under a transmission electron microscope (JEM-1010, JEOL).

Terminal deoxynucleotidyl transferase-dUTP nick-end labeling (TUNEL)

Apoptotic cells were identified using a TUNEL BrightRed Apoptosis Detection Kit (Vazyme, Nanjing, China), as described in our previous study (Gao et al., 2020). In short, sections were deparaffinized, rehydrated, and incubated with proteinase K for 20 min at RT. The slides were then treated with equilibration buffer for 1 h before labeling with BrightRed Labeling Buffer for 1 h at 37 °C. The sections were then washed twice with phosphate-buffered saline (PBS), and stained with Hoechst for 5 min at RT to prepare mounting.

RNA extraction and Real-time quantitative PCR

Total RNA was extracted from the testicular tissues using TRIzol reagent (Vazyme, Nanjing, China), according to the manufacturer’s instructions. RNA was reverse-transcribed into cDNA using a PrimeScript Reverse Transcription Mix (Vazyme, Nanjing, China). Thereafter, cDNA was then analyzed by SYBR Green-based real-time quantitative PCR in an Applied Biosystems 7500 real-time PCR system (Applied Biosystems, Foster City, CA, USA) with 18S rRNA as an internal control. The primers used are as follows: β-catenin, forward 5′-GGCGGCCGCGAGGTA -3′ and reverse 5′- GTGGCTGACAGCAGCTTTTC -3′; Espin, forward 5′- CTTTGGAGCTGGGCAGTTGA -3′ and reverse 5′- TTGAAAGATTTGGTGCTGGGT -3′; Palladin, forward 5′-GCTGGATGTCTACATTTCCCGA -3′ and reverse 5′-CCAGCCAGCCTAAGAAACCA -3′; 18sRNA, forward 5′- AAACGGCTACCACATCCAAG -3′ and reverse 5′- CCTCCAATGGATCCTCGTTA -3′.

Statistical analysis

Data were presented as the means ± SD from at least three replicates for each experiment. The differences between Rai14 WT and KO mice were calculated using unpaired Student’st-test with statistical significance set at ap value of <0.05. The differences among the WT, KO and heterozygous were calculated using one-way ANOVA significance set at ap value of <0.05. Microsoft Excel or GraphPad Prism 6.0 software were used for the statistical analyses.

Results

Generation of Rai14 KO mice

To investigate the physiological function of Rai14, we generated Rai14 global KO mice using CRISPR/Cas9 technology. Two gRNAs were designed to target gene sites in exon 3 of the Rai14 gene, resulting in a 56-bp deletion of exon 3 (Fig. 1A). PCR amplification was performed to rapidly identify the genotypes from Rai14 WT (+/+), KO (-/-), and heterozygous (+/-) mice (Fig. 1B). Furthermore, western blot and immunofluorescence analyses were carried out to evaluate the absence of RAI14 at the protein level (Figs. 1C and 1D). As shown in Fig. 1C, western blot analysis could not detect RAI14 or truncated RAI14 in Rai14- KO testis. Additionally, immunofluorescence staining showed specific distribution of RAI14 in the cytoplasm of elongating spermatids in Rai14 WT testis, whereas no such obvious signal was observed in spermatids of Rai14 KO testis (Fig. 1D). In one previous in vivo study in rats, RAI14 was found to be distributed at the ES junction in rat testis; however, in our study, RAI14 was not found at the ES junction in the testis of Rai14 WT mice (Fig. 1D).

Figure 1 Generation of Rai14−∕− mice.

(A) Schematic diagram of CRISPR/Cas9-mediated Rai14 editing; (B) PCR amplification of genomic DNA in Rai14+∕+,−∕− and+∕− mice; (C) Western blot analysis of RAI14 in Rai14+∕+ and −∕− testes; (D) Co-immunostaining of RAI14 and PNA in Rai14+∕+ and−∕− testes. The epithelial cycle is divided into 12 stages recognized by PNA-labeled acrosomes. RAI14 is specifically located in spermatids at steps 11–14. Rai14−∕− tubules are used as the negative control. Scale bar: 20 µm.

Normal fertility and sperm parameter in Rai14-KO mice

Rai14 KO mice were viable and exhibited normal development. A 4-month-long fertility test revealed that Rai14 KO adult males had normal fertility (Fig. 2A). The testicular weight of Rai14 KO and WT mice were comparable (Figs. 2B and 2C). Moreover, Rai14 KO males exhibited normal sperm concentration, motility, and morphology, compared with WT males (Figs. 2D–2H).

Figure 2 Rai14−∕− mice are fertile.

(A) Fertility test of Rai14+∕+,−∕− and +∕− males. For Rai14+∕+, n = 6; for Rai14+∕−, n = 7, for Rai14−∕−, n = 7 , P > 0.05; (B) Testes of Rai14+∕+ and −∕− mice; (C) Testis/body weight, n = 3, P > 0.05; (D) Sperm concentration in Rai14+∕+ and−∕− mice, n = 5, P > 0.05; (E) Sperm motility in Rai14 +∕+ and −∕− mice, n = 6, P > 0.05; (F) Sperm abnormality in Rai14+∕+ and −∕− mice, n = 4, P > 0.05; (G) HE staining of cauda epididymal sperm from Rai14+∕+ and −∕− mice. Scale bar: 50 µm; (H) Ultrastructural analysis of cauda epididymal sperm from Rai14+∕+ and −∕− mice. Note the normal head and axoneme with typical “9 + 2” microtubule structure (nine pairs of peripheral and two central microtubules, arrows) in Rai14+∕+ and −∕− mice. Nu, nucleus; Ac, acrosome.

Complete spermatogenesis in Rai14 KO testis

Histological analysis of Rai14 WT and KO mice revealed that the morphology of the testis and epididymis of Rai14 KO and WT male mice was indistinguishable from each other (Figs. 3A and 3B). Similar conclusions were drawn on the basis of the normal expression and counts of the spermatogonial stem cell maker Lin28 (Rode et al., 2018) (Figs. 3C and 3D); the spermatocyte marker γH2AX (Wang et al., 2016) (Figs. 3E and 3F); Sertoli cell marker Vimentin (Alsemeh, Samak & El-Fatah, 2020) (Figs. 3G and 3H); and the Leydig cell marker HSD-3β (Cen et al., 2020) (Figs. 3I and 3J). Moreover, based on the results of the TUNEL analysis in our study, the number of apoptotic cells showed no significant difference between the two groups (Figs. 3K and 3L). Altogether, our results strongly demonstrate that Rai14 is not essential for spermatogenesis or fertility in male mice.

Figure 3 Normal spermatogenesis in Rai14−∕− mice.

(A) Periodic Acid Schiff (PAS) staining of testicular sections from Rai14+∕+ and −∕− mice. The epithelial cycle is divided into 12 stages recognized by PAS, according to changes of the acrosome and nuclear morphology of spermatids. Scale bar: 20 µm; (B) HE staining of the cauda epididymis obtained from Rai14+∕+ and −∕− mice. Scale bar: 100 µm; (C) Immunostaining of Lin28 from Rai14+∕+ and −∕− testes; (D) Quantification of (C), n = 5, P > 0.05. Thirty tubules were counted per sample. Scale bar: 20 µm; (E) Immunostaining of γ H2AX from Rai14+∕+ and −∕− testes; (F) Quantification of (E), for Rai14+∕+, n = 5; for Rai14−∕−, n = 4; P > 0.05. Thirty tubules were counted per sample. Scale bar: 20 µm; (G) Immunostaining of Vimentin from Rai14+∕+ and −∕− testes; (H) Quantification of (G), n = 5, P > 0.05. Thirty tubules were counted per sample. Scale bar: 20 µm; (I) Immunostaining of HSD-3 β from Rai14+∕+ and −∕− testes; (J) Quantification of (I), n = 3, P > 0.05. Three slides were counted per sample. Scale bar: 50 µm; (K) TUNEL assay of Rai14+∕+ and −∕− testes; (L) Quantification of (K), n = 4, P > 0.05. Thirty tubules were counted per sample. Scale bar: 20 µm.

ES junction is not disturbed in Rai14 -KO testis

As RAI14 showed highest localization at the ES junction in adult rat testis, and siRNA-mediated Rai14 KD led to the mis-localization of ES-associated proteins (Qian, Mruk & Cheng, 2013a), we sought to assess the localization of basal ES (β-catenin and Espin) and apical ES (Espin and Palladin) proteins (Mruk & Cheng, 2004; Qian et al., 2013b) in both Rai14 WT and KO testes. Both real-time quantitative PCR (Figs. 4A–4C) and immunofluorescence (Figs. 4D–4I) analyses revealed no measurable alterations in the transcript or protein levels of ES-associated genes between the two groups.

Figure 4 Expression and distribution of ES-associated genes/proteins.

Real-time quantitative PCR analysis of β-catenin (A), Espin (B) and Palladin (C) from Rai14+∕+ and −∕− testes. n = 3, P > 0.05; Immunostaining of basal ES protein β-catenin (D, E) and Espin (F, G) from Rai14 +∕+ and −∕− testes. Scale bar: 20 µm; Immunostaining of apical ES protein Espin (F, G) and Palladin (H, I) from Rai14+∕+ and −∕− testes. Scale bar: 20 µm.

Rai14 is not required for F-actin organization in mouse testis

As an actin-binding protein, RAI14 participates in F-actin organization at the ES junction in rat testis (Qian, Mruk & Cheng, 2013a). Here, we used phalloidin-labeled F-actin staining to observe actin filaments surrounding the heads of elongating spermatids. In both Rai14 WT and KO testes, the actin filament bundles were intact and organized so that they tightly surrounded the spermatid heads (Fig. 5A). Furthermore, transmission electron microscopy (TEM) of the apical ES was performed in both the groups to better visualize the actin bundle organization. Ultrastructurally, the apical ES junctions in both groups consisted of actin filaments bundles sandwiched between the cisternae of the endoplasmic reticulum and the apposed Sertoli-spermatid plasma membranes (Fig. 5B). These data indicate that Rai14 is not essential for the assembly of actin filaments at the apical ES in mouse testis.

Figure 5 Rai14 is not required for F-actin organization.

(A) Phalloidin-labeled F-actin staining of Rai14+∕+ and −∕− spermatids at steps 13–14. Scale bar: 20 µm; (B) TEM analysis of the apical ES from Rai14+∕+ and −∕− spermatids at steps 13–14. ES synchronously stretches along with the acrosome, and is characterized by the presence of actin filament bundles (arrows). Nu, nucleus; Ac, acrosome; ER, endoplasmic reticulum.

Discussion

Previous microarray analyses have identified over 2,300 genes that are enriched in male germ cells (Schultz, Hamra & Garbers, 2003). Thereafter, many studies have been performed to characterize testis-enriched genes/proteins based on transcriptomics and proteomics (Bonilla & Xu, 2008; Clement et al., 2007; Djureinovic et al., 2014; Pineau et al., 2019; Uhlen et al., 2015). In addition to housekeeping genes, testis-enriched genes have, for a long time, been thought to play a crucial role in spermatogenesis. However, using gene-KO approaches, Miyata et al. have revealed 54 testis-enriched genes that are dispensable for male fertility in mice (Miyata et al., 2016). Since then, several studies have established a number of KO mice models without obvious fertility phenotypes (Feng et al., 2018; Holcomb et al., 2020; Lu et al., 2019; Park et al., 2020; Wang, Chen & Liu, 2018a; Wang et al., 2018b; Zhang et al., 2019). Similarly, we used CRISPR/Cas9-based gene editing in our study and identified Rai14, which was enriched in the testis and was dispensable for spermatogenesis and fertility in mice. Considering these findings, we believed that the phenotypic information obtained in our study can inform other researchers and prevent them from conducting redundant experiments. Moreover, these results can serve as a basic resource for genetics studies on human fertility.

RAI14 has been previously considered as an actin cytoskeleton-associated protein purified from rat liver tissue (Peng et al., 2000). Several studies have revealed that RAI14 is expressed in various tissues and cells, but is highly expressed in both human and mouse testes (Kutty et al., 2001; Kutty et al., 2006b; Yuan et al., 2005). In rat testis, RAI14 was found to be exclusively located at the ES junction, most abundantly at the apical ES. SiRNA-based Rai14 KD in rat testis led to defects in elongating spermatid polarity and transport, and finally caused spermiation failure. Mechanistically, RAI14 physiologically interacts with actin and another actin cross-linking protein, Palladin. As suggested in previous studies, the altered phenotype caused by the loss of RAI14 may be associated with the mis-localization of F-actin and Palladin at the apical ES (Qian, Mruk & Cheng, 2013a; Qian et al., 2013b). However, in this study, RAI14 distribution occurred specifically in the cytoplasm of elongating spermatids, but not at the ES. Meanwhile, RAI14 fluorescence signals were undetectable in Rai14 KO testis, further supporting the specificity of the antibody against RAI14. Furthermore, Rai14 KO mice displayed normal spermatogenesis and fertility. Histological analysis revealed no difference in the ES structure, actin filament bundle organization, as well as ES associated protein distribution between the two groups. Thus, the question of the reasons for the phenotypic differences in KD versus KO arises. In our opinion, at least two possibilities contribute to them. First, the different distributions of RAI14 in rat and mouse testis suggest that RAI14 plays different roles in various species. In addition, these phenotypic differences could also be explained by functional compensation from paralogs in KO model or off-target effects in KD.

Conclusions

In summary, we achieved Rai14 global KO mice by using Cas9/sgRNA-mediated gene editing. Our results provide proof-of-principle evidence to show that Rai14 is neither required for the ES junction nor spermatogenesis in mice.

Supplemental Information

Supplemental Information 1 Raw data of Fig. 2A

Fertility test.

Click here for additional data file.

Supplemental Information 2 Raw data of Fig. 2C

Weight/body.

Click here for additional data file.

Supplemental Information 3 Raw data of Figs. 2D–2F

Sperm parameters.

Click here for additional data file.

Supplemental Information 4 Raw data of Figs. 3C–3G

Cell counts.

Click here for additional data file.

Supplemental Information 5 Raw data of Figs. 4A–4C

Transcriptional levels of genes.

Click here for additional data file.

Supplemental Information 6 Gels and Blots of Figs. 1B–1C

Full-length uncropped gels and blots.

Click here for additional data file.

Supplemental Information 7 Check list

Click here for additional data file.

Additional Information and Declarations

Competing Interests

Author Contributions

Animal Ethics

Data Availability

The authors declare there are no competing interests.

Yangyang Wu performed the experiments, analyzed the data, prepared figures and/or tables, authored or reviewed drafts of the paper, and approved the final draft.

Ting Wang, Zigao Zhao and Siyu Liu performed the experiments, analyzed the data, prepared figures and/or tables, and approved the final draft.

Cong Shen, Hong Li and Mingxi Liu performed the experiments, prepared figures and/or tables, and approved the final draft.

Bo Zheng conceived and designed the experiments, prepared figures and/or tables, authored or reviewed drafts of the paper, and approved the final draft.

Jun Yu and Xiaoyan Huang conceived and designed the experiments, authored or reviewed drafts of the paper, and approved the final draft.

The following information was supplied relating to ethical approvals (i.e., approving body and any reference numbers):

The Animal Ethical and Welfare Committee (AEWC) of Nanjing Medical University approved this research (Approval No IACUC-2004020).

The following information was supplied regarding data availability:

Raw data are available as Supplemental Files.

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
