# Peer review of "Retinoic Acid Induced Protein 14 (Rai14) is dispensable for mouse spermatogenesis"

_PeerJ, doi:10.7717/peerj.10847_

## Round 0.1 · original submission · Major Revisions

You will see that both reviewers generally find the results interesting, but have some concerns about the figures and methodological issues which can be addressed in re-review.

The manuscript would benefit from significant editing to improve the grammar and Reviewer 2 has pointed out a number of issues that should be addressed.

Reviewer 1 ·

Basic reporting

The manuscript describes the knockout of Rai14 in mouse model and it's effects on spermatogenesis. The conclusions of this manuscript show that Rai14 is dispensable for mouse spermatogenesis, which is a different result from a rat model. The manuscript could benefit from a more thorough background and/or conclusions section. For example, what could be the reasons for the differences in the different animal models? Are there differences in the protein function? How is this applicable to human health and male fertility?
The manuscript is generally well written and clear, but should be edited one more time for English language. E.g. line 66 - among the various junctions existed in the seminiferous...and line 79 - Theses domains.
The raw data for Figure 2C shows body weight for 3 animals and one testicular weight for each animal. Was only one testis weighed?
The fluorescent images are very hard to see. In 1D, it may be helpful to draw a line around the tubule, it's hard to discern what's what in that image. Same for 3 C-G. Also, in 3 C-G, it would be helpful to see less magnified images to get an idea of the expression pattern of each marker. Figure 4 D-G, it is very hard to see what the authors are trying to show. Again, in addition to the current images, a less magnified image would be helpful.
What is the purpose of Figure 2 G? It is hard to see the actually morphology of the sperm with this magnification.

Experimental design

The study is well designed and the results support the conclusions. However, can the authors add for Figure 3 C-G, how many samples were analyzed, were cross sections counted, if yes, how many cross-sections per group.

Validity of the findings

The results are valid and supported by the data. All raw data files are provided.

Reviewer 2 ·

Basic reporting

Grammar needs to be improved.

Experimental design

See general comments, below.

Validity of the findings

See general comments, below.

Additional comments

Grammar needs to be improved throughout, perhaps through the assistance of an English-speaking colleague or grammar service.

64-6: would not say spermatogenesis occurs as a result of “extensive restructuring” – these things occur in somatic cells, but there is no evidence these direct spermatogenesis as implied here
67-73: this reviewer’s understanding is that the apical and basal ESs are distinct junctions, both in location and construction (basal = tight junction, and apical = adherens junction?), and therefore likely should not be grouped and described as if they are one and the same; please clarify
75: “proved to be”? That’s a strong statement – what evidence is provided outside of RPE cells? In the testis? And it’s obviously regulated by more than just RA, if it’s only expressed in the testis and placenta in vivo
79: replace “generally”
80-81: “located at both Sertoli and germ cells” – what does this mean? In all germ cells? In all Sertoli cells?
82: replace “could” – is this just speculation? Do they actually do it, or just when they feel like it?
83: typically siRNA approaches don’t “silence”, but knock down mRNA levels; and what age Sertoli cells were used, were they adult, or juvenile before these junctions formed? And if in vitro, were these junctions formed? If not, how useful are those observations?
86: “impeded” means slow down, delay, hinder… none of these really sound like they describe spermatid polarity and adhesion; what is meant by “spermatid movement”? Is this the observation that spermatids tend to be found at different points in the epithelium during spermiogenesis? Is this critical for their development? What mechanisms are known to regulate it?
89: replace “several lines of evidence” with something like – it has been observed in several studies that there are phenotypic differences in KD vs KO models
95-6: Seems strange that the entire Introduction is leading the reader to believe that RAI14 has an important role, and then the conclusion comes as a let-down – which it shouldn’t, the data is the data… would consider re-framing the Introduction so this isn’t so ‘jarring’
100: is “healthy” really necessary? Does anybody working in reproduction purposefully use unhealthy mice?
Methods: Describe how was fixation done for IF, also electron microscopy
155: sperm from the cauda epididymis are far from “mature”
155-6: using what software, and what parameters? More details are required
160: epididymides = plural
203: replace “elimination” with absence
208: was that previous result in testis in vivo, or in Sertoli cell cultures? Please clarify
223: was this gammaH2AX or phosphorylated gammaH2AX?
224: typo
230: but not in your observation, correct? And it wasn’t expression, but rather localization
241-2: how did the authors distinguish the phalloidin staining for F-actin in the apical ES vs the tubulobulbar complex, a spermatid F-actin rich structure?
251-2: A rather odd non sequitur to start the Discussion – and lots and lots of studies using bulk and single cell RNA-seq have been done since that 2003 study; and the Ikawa and Matzuk labs have published several other papers more recently, but with the same conclusion that could be cited here

280: why use “efficient Rai14 targeting” to describe a global KO? It was all-or-nothing

Fig. 1 – Negative control for immunostaining (in D)? What specifically is RAI14 localized to? What stage tubule is shown in each? Does RAI14 localization differ during spermiogenesis? Provide a higher-mag image that shows staining more clearly.

Figures – the different figures, symbols, centered, etc need to be lined up, equivalent spacing provided

Fig. 5 – What stages?

---

## Round 0.2 · accepted · Accept

Congratulations! Both reviewers found your revisions addressed all of your concerns and recommended it for publication. I commend you on tackling their concerns head on in a timely manner and look forward to seeing it online.

Reviewer 1 ·

Basic reporting

The authors have made the requested changes and considerably improved the quality of the manuscript.

Experimental design

Experimental design is adequate and well defined.

Validity of the findings

Results are valid.

Reviewer 2 ·

Basic reporting

Manuscript is much improved in terms of readability and data presentation.

Experimental design

No comment.

Validity of the findings

No comment.

Additional comments

The authors should be commended for taking the reviewers' suggestions seriously and addressing each point. It is this reviewer's opinion that the manuscript is much improved.